# Lepidopteran wing scales contain abundant cross-linked film-forming histidine-rich cuticular proteins

Jianqiu Liu[1,2], Zhiwei Chen[1], Yingdan Xiao[1,2], Tsunaki Asano[3], Shenglong Li[1], Li Peng[4], Enxiang Chen[1,2], Jiwei Zhang[1], Wanshun Li[1,2], Yan Zhang[1,2], Xiaoling Tong[2], Keiko Kadono-Okuda[5], Ping Zhao[2], Ningjia He[1], Kallare P. Arunkumar[1,2,6], Karumathil P. Gopinathan[7], Qingyou Xia[2], Judith H. Willis [8], Marian R. Goldsmith [1,2,9✉] & Kazuei Mita [1,2✉]

Scales are symbolic characteristic of Lepidoptera; however, nothing is known about the contribution of cuticular proteins (CPs) to the complex patterning of lepidopteran scales. This is because scales are resistant to solubilization, thus hindering molecular studies. Here we succeeded in dissolving developing wing scales from *Bombyx mori*, allowing analysis of their protein composition. We identified a distinctive class of histidine rich (His-rich) CPs (6%–45%) from developing lepidopteran scales by LC-MS/MS. Functional studies using RNAi revealed CPs with different histidine content play distinct and critical roles in constructing the microstructure of the scale surface. Moreover, we successfully synthesized films in vitro by crosslinking a 45% His-rich CP (BmorCPR152) with laccase2 using N-acetyl- dopamine or N-β-alanyl-dopamine as the substrate. This molecular study of scales provides fundamental information about how such a fine microstructure is constructed and insights into the potential application of CPs as new biomaterials.

[1] State Key Laboratory of Silkworm Genome Biology, Southwest University, Chongqing, China. [2] Biological Science Research Center, Southwest University, Chongqing, China. [3] Department of Biological Sciences, Tokyo Metropolitan University, Tokyo, Japan. [4] Shanghai Center for Plant Stress Biology and Center of Excellence in Molecular Plant Sciences, Chinese Academy of Sciences, Shanghai, China. [5] Institute of Agrobiological Sciences, National Agriculture and Food Research Organization (NARO), Tsukuba, Japan. [6] Central Muga Eri Research and Training Institute, (CMER&TI), Central Silk Board, Jorhat, India. [7] Indian Institute of Science, Bangalore, India. [8] Department of Cellular Biology, University of Georgia, Athens, GA, USA. [9] Department of Biological Sciences, University of Rhode Island, Kingston, RI, USA. ✉email: mki101@uri.edu; mitakazuei@gmail.com

Scales are the most remarkable and symbolic characteristics of Lepidoptera. The brilliant and patterned colors underwritten by scales play important roles in mating behavior[1], camouflage and warning coloration[2–4], temperature regulation[5–7], and protection against predators[8,9]. Butterfly wing scale geometry and surface patterning are also reported to improve flight efficiency[10]. Having scales that cover the wings and body of butterflies and moths increases their chances of survival and reproduction in diverse environments.

A single layer of epidermal cells (resting, as always, on a sheet of basement membrane) is responsible for producing and shaping the cuticle. Epidermal cells can undergo a series of divisions culminating in a final differential division that yields a scale and a socket cell[11–13]. Cuticular proteins (CPs) and chitin synthesized de novo by scale cells accumulate around them and gradually form very hard scales[13,14]. Varied types of scales are precisely arranged in different regions of wings and body to produce unique color patterns[15].

Having various shapes from cylindrical to flat, the scale is basically an envelope with "upper" and "lower" surfaces. The standard (unspecialized) scale is typically characterized by a series of ridges, crossribs, lamellae, microribs, and pigment[14,16]. Differently colored lepidopteran wing scales have different cuticular micro structures[17–19], and pigmentation of scales[20,21] coincides with the development of their fine morphological features[16]. A study in the Tiger Swallowtail *Papilio glaucus* first suggested that sclerotization is linked to final scale color[20,22,23]. Matsuoka and Monteiro reported that melanin pathway genes play a dual role in scale construction and pigmentation by knockout of the pigment enzymes in the butterfly *Bicyclus anynana*[24]. Histochemical studies of butterfly wing scales show that F-actin plays an essential role in scale elongation and ridge position in early development[25]. Recent findings show F-actin is also important for the morphogenesis of swallowtail butterfly wing scale nanostructures[26]. However, the involvement of CPs per se in the formation of the intricately layered cuticular microstructure with other associated proteins remains unclear[14,27,28]. Although an earlier proteomic analysis tried to identify the proteins in mature scales from *Bombyx mori* adults, only seven CPs were detected due to their insolubility[29].

In this study, we succeeded in identifying many CPs and other classes of proteins by collecting scales from developing wings before crosslinking occurred and dissolving them in a buffer containing urea, a well-established method for studying cuticle CPs but a breakthrough approach for the molecular study of wing scales. LC-MS/MS of the soluble scale proteins allowed us to identify the CPs and enzymes related to cuticular sclerotization, which revealed a possible mechanism of scale formation. RNAi experiments conducted with histidine-rich (His-rich) CPs with different His-contents suggested that each His-rich CP identified in scales might play a distinct, but vital role in scale microstructure. Films produced in vitro by crosslinking reactions of the highest His-rich CP (45%; BmorCPR152) suggested a possible avenue to industrial application of CPs as new biomaterials.

## Results and discussion

### The development of scale microstructures

Morphological observation of *B. mori* wing scales revealed vivid and dynamic development during the pupal and pharate adult stages (Fig. 1a). As reported for *Precis coenia*, scales emerged from their sockets by 72 h after pupation[12]. In *B. mori* (upper panel of Fig. 1a), tiny round scales were clearly visible at the third day after pupation (P-3), and then grew to an elongated baseball bat-like shape in P-4. The size and shape of scales in P-5 became similar to the mature ones; however, they remained soft and even dissolved partially in phosphate buffered saline (PBS). Scales in P-6 retained their shape very well in PBS although they still were soft. High magnification images of scales by scanning electron microscopy (SEM) showed that the ridge and cross rib framework was already present at P-5 (lower panel of Fig. 1a). Lamellae showed clearly and the ridges became more complicated and three-dimensional at P-6. All the observations indicated that P-6 scales, which have the same microstructures as mature scales, could be dissolved in urea containing buffer, since the stabilization of the microstructure of the scales began at P-6 or later stages.

### The dissolving of developing scales in buffer

The developing wing scales were collected from the sixth and eighth days after pupation, followed by dissolving in 8 M urea-containing PBS and SDT buffer (4% (w/v) SDS, 100 mM Tris/HCl, PH ~7.6) as described in Fig. 1b and Supplementary Fig. 1. The SDS-PAGE gels of proteins extracted from solubilized P-6 scales presented clear and discrete bands, whereas very faint bands appeared in P-8 (Fig. 1c). These results showed that scales of P-8 became much more difficult to dissolve compared to P-6 scales, which was consistent with our morphological observations indicating that the scales started to stabilize around P-6. As the proteins and other components were stabilized, the number and amount of proteins extractable from scales were also expected to be reduced.

### Identification of proteins involved in scales

We carried out liquid chromatograph-tandem mass spectrometry (LC-MS/MS) analysis of extracts from P-6 and P-8 wing scales which detected peptides (Supplementary Data 1–4) originating from 1257 and 1090 proteins. The proteins classified as unreliable are indicated by "U" in column G of Supplementary Data 1–4. Only the reliable proteins were used for the analysis in this report; however, the unreliable proteins may also contain useful information.

Our analyses also provided quantitative data as iBAQ (intensity-based absolute quantification) scores for all of the identified proteins. Table 1 shows the top 25 scores for the two replicates of each time point; complete data are listed in Supplementary Data 1–4. What is immediately obvious is that CPs were among the most abundant proteins identified. In the two P-6 samples, they were 68% of the top 25, while at P-8 they were 64% and 56% in replicate samples. These most abundant proteins belonged to the CPG family (CPs with glycine-rich repeats), CPLCP (CPs with low complexity proline-rich amino acids, and also characterized by three conserved sequence motifs[30,31]), CPCFC (CPs with well-conserved cysteine residues), CPFL (containing a 44 amino acid motif-like domain in the conserved C-terminal region), CPAP3 and CPR family (CPs with the Rebers and Riddiford Consensus, which has been subdivided into three groups, RR-1, RR-2 and RR-3). There were also three different CPHs (a collection of unclassified CPs). Obviously, scale formation involves a complex selection of CPs to build the exquisite scale structure. This finding is reminiscent of an earlier LC-MS/MS study that identified members of several CP families in the corneal lens and other structures from the mosquito *Anopheles gambiae*[32].

Interestingly, 23 of the identified CPs from P-6 and 20 from P-8 were His-rich, with histidine contents amounting to 6-45%. More peptides with His-residues were detected in P-6 compared to P-8 (Supplementary Figs. 2 and 3), and His-rich RR-2s had obvious predominance in both P-6 and P-8 scales (Fig. 1d). Iconomidou et al. (2005) published a homology model for RR-2 proteins showing that the His residues in the Consensus region were in positions, where they could participate in cross-links[33]. Recently, Liu et al. reported[34] that *B. mori* has 73 His-rich CPs

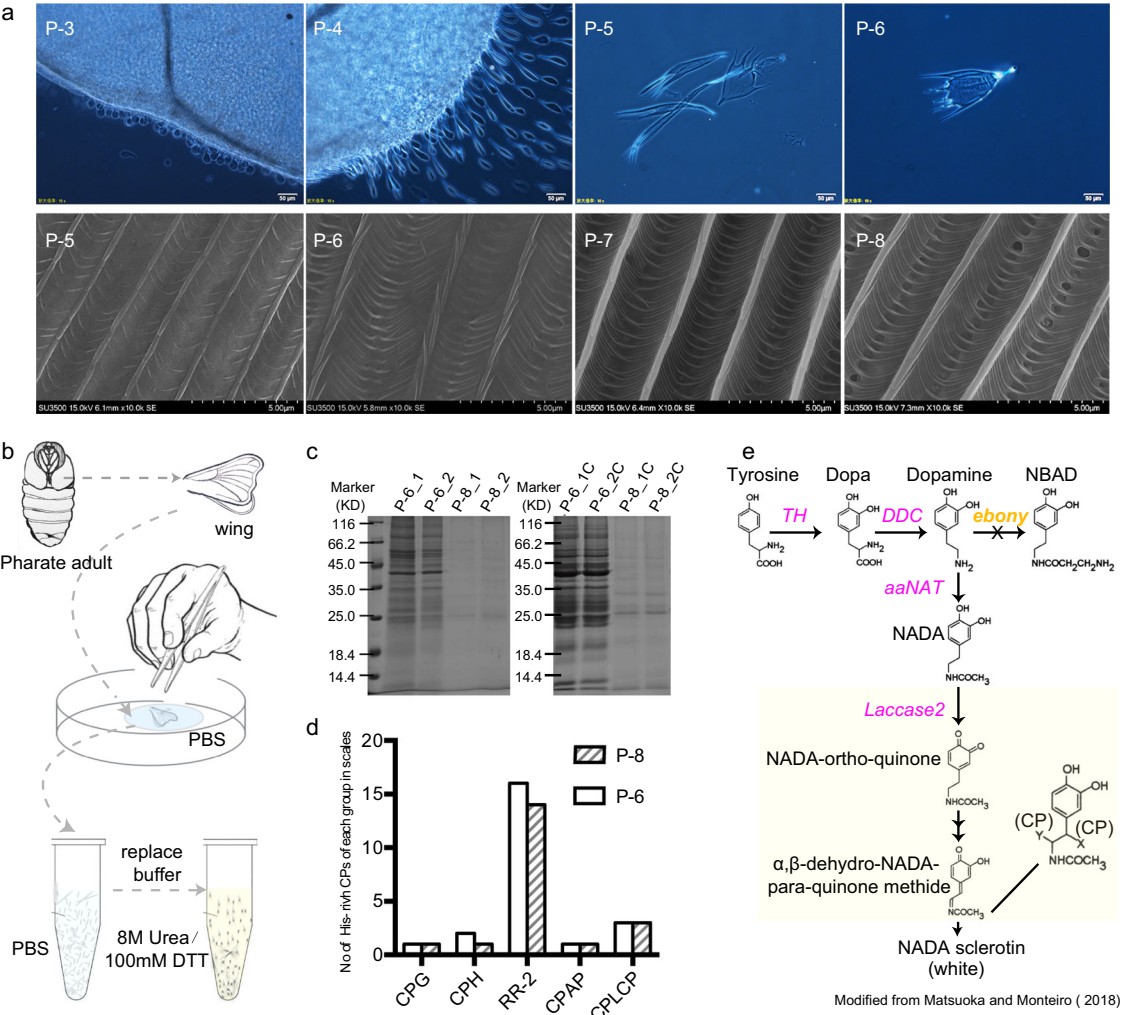

**Fig. 1 Development, structural components, and related enzymes of wing scales. a** Development of wing scales. Upper panels: light microscopic images showing development of scales on pupal/pharate adult wings. Scale bar = 50 μm; Lower panels: scanning electron microscopic images of isolated wing scales at various pupal/pharate adult stages. Scale bar = 5 μm. **b** Schematic diagram of the isolation of CPs in pupal wing scales of P-6 and P-8 pupae. The detailed procedure is presented in Supplementary Fig. 1. **c** SDS-PAGE gel of proteins extracted from isolated wing scales of P-6 and P-8. Duplicate wing scale samples were solubilized first in 8 M Urea/100 mM DTT, and then proteins from pellets were solubilized in SDT buffer with ultrasonic treatment. Lanes: P-6/8-1 and P-6/8-2, supernatant of P-6/8 wing scales solubilized in 8 M urea/100 mM DTT; P-6/8-1C and P-6/8-2C, pellets from P-6/8 wing scales from first treatment, followed by suspension in SDT buffer and ultrasonic treatment. **d** Number of His-rich CPs in each CP group from wing scales in P-6 and P-8. **e** Cuticular sclerotization pathway. Enzymes identified by LC-MS/MS are colored magenta. Ebony (yellow) was not detected in the wing scales of *B. mori*.

most belonging to the RR-2 group, and hypothesized that His-rich CPs may contribute to construct tough cuticle structure[34]. Our findings of abundant His-rich RR-2 CPs in developing wing scales are consistent with the hypothesis of Liu et al., that His-rich CPs participate in cuticular sclerotization via the imidazole group of His residues reacting with the quinones[34]. Compared to other his-rich RR-2 proteins, BmorCPR57 (15% His-content) was far more abundant in the recovered proteins of both P-6 and P-8 (Table 1). The highest and second highest His-content proteins BmorCPR152 (45%) and BmorCPR82 (27%) were also found in the top 25 (Table 1).

Among the identified proteins, we reclassified three glycine-rich CPs (CPG) that were initially classified by Futahashi[35] as CPLCPs. Based on BlastP searches against CPLCPs of *An. gambiae*, we renamed BmorCPG12, 13, and 24 as BmorCPLCP12, 11, and 3. Those were also His-rich (6–8%) and in high amounts in scales (Table 1 and Supplementary Data 1–4). Fu et al. previously reported[29] that BmorCPLCP12 is found in the mature

body scales of *B. mori*. Moreover, BmorCPLCP11 and 12 are homologs of the *An. gambiae* CPs AgamCPLCP11 and 12, which previous proteomics analysis found to be present in larval head capsule and cast pupal cuticles[36,37]. Corman and Willis[38] found that the mosquito CPLCP family is similar to articulins, which are intracellular proteins of protists that form ordered, filamentous structures and have been suggested to provide support, patterning, and some elasticity to the cytoskeleton[39]. Accordingly, CPLCPs may also play an important roles in sculpting the basic structure of lepidopteran scales.

Gene 011414 was also among the most abundant proteins recovered, especially at P-8. PFAM classifies this protein as a retinin. The retinin domain was identified in the CPCLA family of *An. gambiae*, but the *B. mori* protein did not score as a CPLCA. Nonetheless, based on its abundance, it is probably another CP, underscoring the complexity of scale construction. Also among the 25 most abundant proteins recovered were apolipoporins, lipoproteins, two different sensory proteins, and one with the von

**Table 1 Rank order of the most abundant proteins recovered based on iBAQ scores (data from Supplementary Data 1–4).**

| Rank order | P6-1 | P6-2 | P8-1 | P8-2 |
|---|---|---|---|---|
| 1 | **CPLCP12** | CPG31 | CPG31 | CPG31 |
| 2 | CPG31 | **CPLCP12** | CPG41 | Retinin |
| 3 | **CPR57** | **CPR57** | Gene003538 | Gene003538 |
| 4 | **CPLCP11** | **CPLCP11** | Retinin | Gene009269 |
| 5 | **CPH30** | **CPH30** | Gene009269 | CPCFC |
| 6 | **CPR91** | **CPG9** | CPCFC | CPR135 |
| 7 | CPG41 | CPG41 | CPR135 | **CPR57** |
| 8 | **CPG9** | **CPR91** | **CPR57** | CPG41 |
| 9 | CPR137 | CPG38 | CPG38 | Gene013570 |
| 10 | Retinin | CPR135 | CPFL3 | Gene011501 |
| 11 | CPR135 | Retinin | Gene013570 | CPH4 |
| 12 | **CPLCP3** | **CPLCP3** | Gene011131 | **CPR91** |
| 13 | Gene009150 | Gene009150 | CPF | Gene011131 |
| 14 | **CPR82** | Gene010574 | **CPR74** | Gene010938 |
| 15 | CPG38 | Gene003746 | Gene011501 | CPG38 |
| 16 | Gene010574 | CPG38 | Gene010938 | CPFL3-2 |
| 17 | **CPR126** | CPR48 | CPH4 | Gene016128 |
| 18 | Gene003746 | **CPR82** | CPT3 | CPR137 |
| 19 | **CPR128** | **CPR126** | CPR137 | **CPH30** |
| 20 | Actin | Gene009460 | CPG20 | **CPR152** |
| 21 | CPR48 | Gene003538 | **CPLCP12** | **CPLCP12** |
| 22 | Gene011501 | Gene011506 | CPR73 | Actin |
| 23 | Gene009460 | Actin | **CPH30** | Gene009460 |
| 24 | **CPR133** | **CPR133** | Gene008255 | Gene011502 |
| 25 | Gene003538 | **CPH42** | Actin | CPAP2-A2 |

Abbreviated gene descriptions based on PFAM searches (https://pfam.xfam.org/). The gene names of His-rich CPs are in bold. Additional details are in Supplementary File 2.

Willebrand factor domain, previously reported in the extracellular matrix (matrisome) of Drosophila melanogaster[40].

As previously reported, F-actin and chitin play important roles in the development of scales in butterflies[25,26,41]. We identified actins in the scales, and at both P-6 and P-8 they were among the 25 most abundant proteins (Table 1). Some chitinases, which can cleave internal bonds in chitin polymers[42], were also found in B. mori scales. This is consistent with observations in butterflies that F-actin bundles are degraded once the ridges have grown at approximately 60% of pupal development[26,41]. In addition, a recent report of scale development in a swallowtail butterfly shows that Arp2/3 appears to reorganize the degraded F-actin to a network after the development of ridges is complete. This reorganization contributes to the morphogenesis of wing scale nanostructures[26]. The detection of Arp2/3 (Gene009685) at P-6 (Supplementary Data 1–2) in our study suggested re-organization of F-actin may also occur in B. mori scales. However, we still cannot draw a conclusion for the functions of F-actin and chitin in forming the microstructure of scales, suggesting that how CPs, F-actin and chitin interact to contribute to the scales of Lepidoptera is an important study for the future.

In sum, this comprehensive analysis of the proteins that make up scales provides crucial information for an understanding of the process of scale formation. Notably, it reveals the existence of a critical previously unsuspected role for His-rich cuticle proteins, which are fundamental to achieving the complex structure of lepidopteran scales.

**Identification of cuticular enzymes in scales associated with melanization and sclerotization.** Previously discovered enzymes in the melanin pathway that regulate the intricate morphology of butterfly wing scales[17,22–24] led us to look for evidence of those enzymes in silk moth wing scales. We detected peptides for tyrosine hydroxylase (TH: Gene000277)[43], dopa decarboxylase[44] (DDC: Gene002084), and dopamine N-acetyltransferase[45]

(aaNAT: Gene010727) in developing P-8 scales. These findings suggested that N-acetyl-dopamine (NADA)[46], a sclerotizing precursor molecule along with N-β-alanyl-dopamine (NBAD)[47–49], can be produced from tyrosine (Fig. 1e). However, ebony (NM_001145321.1)[50,51], which is an essential enzyme for the synthesis of NBAD from dopamine, was absent from silk moth scales, which indicated NBAD could not be produced during scale formation.

Not only are NADA and NBAD the final products of the melanin pathway in insects[24], they are also considered the most common precursor molecules in the sclerotization process[48,49]. Andersen suggested a single highly reactive dehydro-NADA or dehydro-NBAD could react with two amino acid residues in different CPs, thereby forming stable crosslinks[48,49,52,53]. That peptides from laccase2A (Gene005466)[54], tyrosinase (Gene009702), and prophenoloxidase (Gene009293)[55] were also identified in scales suggested that dehydro-NADA can be oxidized from NADA, which will lead to NADA sclerotin in scales. The absence of ebony indicates that NBAD sclerotin might not occur in scales. Silk moth scales are mainly white, consistent with the fact that NADA sclerotin is often colorless or very light straw-colored, whereas NBAD products of sclerotization contribute to a yellow color[24,49,53].

His residues, which contain the strongest nucleophilic imidazole groups, have been confirmed to be involved in the sclerotization process in the mealworm (beetle) Tenebrio molitor and the desert locus Schistocerca gregaria[56]. The identification of CPs with a range of His residues together with a series of cuticular enzymes involved in sclerotization suggest the possibility that His-rich CPs have specific roles in the construction of scale microstructure. DDC and aaNAT were only detected in P-8 scales indicating that the pigmentation and sclerotization of wing scales occurred after P-6 or during P-8, before adult eclosion. These observations are consistent with our finding that scales were more soluble in P-6 than P-8. These findings are also consistent with the model proposed by Matsuoka and Monteiro for delayed cross-linking of melanin pathway gene products after the initial

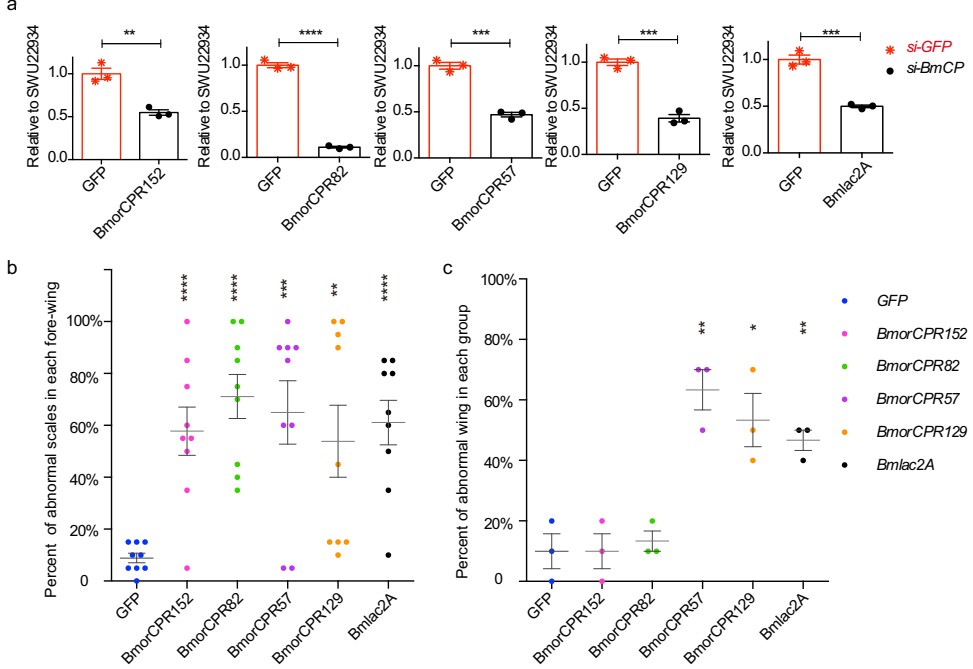

**Fig. 2 Statistical analysis of abnormal wing scales and wing after knockdown of four His-rich *CP* genes and *Bmlac2A* by RNAi. a** Reduction of mRNAs by siRNA confirmed by RT-qPCR ($n = 3$). Red asterisks, control (GFP); black dots, siRNA treated. **b** Percent of abnormal wing scales after RNAi ($n = 9$). **c** Percent of abnormal wings after RNAi ($n = 3$). Blue, GFP control; magenta, BmorCPR152; green, BmorCPR82; purple, BmorCPR57, orange, BmorCPR129, black, Bmlac2A. Significance difference: *$P$ value < 0.05, **$P$ value < 0.01, ***$P$ value < 0.001, and ****$P$ value < 0.0001.

formation of scale ultrastructure in cuticular skeleton development of butterfly wings[24].

**Knockdown experiments of His-rich CPs and Bmlac2A.** Knockdown experiments using siRNA injection were performed to determine the functions of His-rich CPs and the laccase2 enzyme of *B. mori* (Bmlac2A), the key enzyme for sclerotization, in the formation of scale microstructures. The reduction of all of the mRNAs involved in this study was confirmed by RT-qPCR, which indicated reductions of 35–90% relative to controls injected with si-GFP (Fig. 2a and Supplementary Data 5).

Reduction of the mRNAs of the four His-rich *CP*s with different His-content (45, 27, 15, and 10%) and *Bmlac2A* led to abnormal microstructures in over half of the examined scales (details in Fig. 2b and Supplementary Data 5). Compared to normal (Fig. 3a, b), reduction of the mRNA for *BmorCPR152*, which encodes a 45% His-rich CP, produced distorted and bent ridges (magenta arrows in Fig. 3b) in the scale. Reducing mRNA for the 27% His-rich CP *BmorCPR82* rendered a lack of microribs (orange arrow head in Fig. 3b) and disordered crossribs (yellow arrow head in Fig. 3b). After depletion of transcripts for *BmorCPR57* (15% His) and *BmorCPR129* (10% His), the junctions between ridges and crossribs (white arrows in Fig. 3b) were disordered or destroyed and the spacing of ridges became irregular; this damage was especially obvious with *BmorCPR129*. Reduction of *Bmlac2A* mRNA beginning in the pupal stage caused many abnormal microstructures. The cross ribs were intertwined at the window region (yellow arrow heads in Fig. 3b), which resulted in the interruption of the connection between adjacent ridges, and many unexpected striations intersected the microribs (white arrow head in Fig. 3b). The ridges also had abnormal structures (magenta arrows in Fig. 3b); and extra windows were found in the microrib area (blue arrow head in Fig. 3b). All the abnormal phenotypes that were observed from RNAi treatment of individual His-rich *CP* genes were found after

depletion of transcripts of *Bmlac2A*. This similarity in the knockdown results of His-rich *CP* genes and *Bmlac2A* indicated that the processes of cuticular sclerotization played critical roles in making the microstructure of scales.

In addition, the overall development of the wing was affected by reduction of mRNAs of the His-rich *CP*s and *Bmlac2A* compared to control (Fig. 2c, Supplementary Fig. 4, Supplementary Data 5). After siRNA injection of the 15 and 10% His-rich *CP*s, wings were not fully expanded after the moth emerged (Supplementary Fig. 4, rows 2 and 3). Reduction of *Bmlac2A* mRNA produced a more pronounced effect, in that moths had rather small wings (Supplementary Fig. 4, row 4), whereas there was no apparent effect on wing structure from knockdown of the 45% and 27% His-rich CPs (Fig. 2c).

Staining with antibodies against BmorCPR152 (45% His-rich) and BmorCPR82 (27% His-rich) showed they were mainly distributed in the scales (yellow arrows in Fig. 4 rows 1 and 2), whereas BmorCPR57 (15% His-rich) was localized not only in scales (yellow arrows in Fig. 4 row 3), but also in the wing blades (white arrows in Fig. 4 row 3). This is consistent with our finding that RNAi with *BmorCPR152* and *BmorCPR82* caused severe damage only to the structure of the scales, whereas the depression of *BmorCPR57* resulted in more widespread general wing abnormalities despite its lower His-content (Supplementary Fig. 4, row 2).

We also examined the scales from the body and legs after RNA knockdown, which showed abnormalities similar to the microstructure of wing scales (Supplementary Figs. 5 and 6 and Supplementary Data 6). We found severe damage to the compound eyes and mandibular bristles after interference in the expression of the His-rich *CP*s and *Bmlac2A* (Supplementary Fig. 7). Especially, the surface of the compound eyes was shrunken by reduction of the mRNAs of *BmorCPR152* and *Bmlac2A* (Supplementary Fig. 7a). In addition, some mandibular bristles were missing after RNAi treatment with *BmorCPR152*

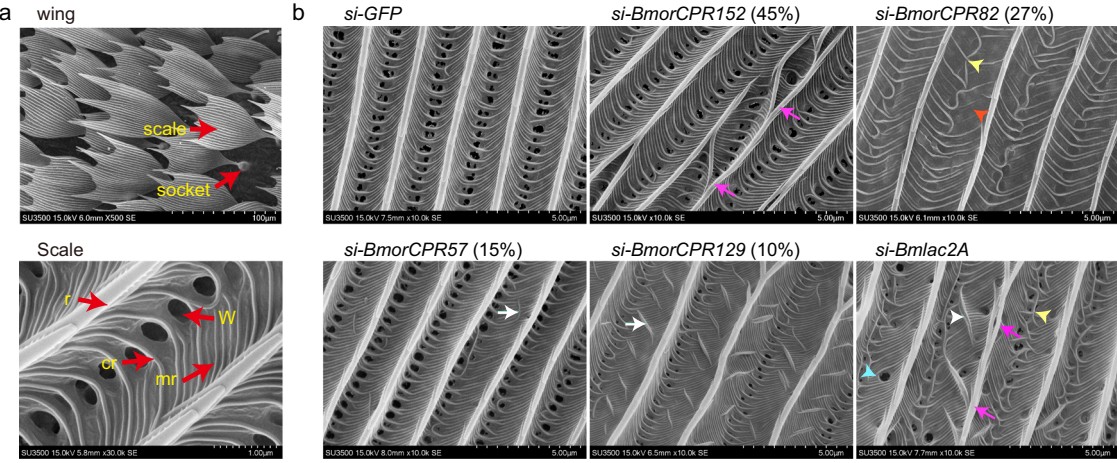

**Fig. 3 Scale microstructure before and after injection of siRNA directed against His-rich *CPs* and *Bmlac2A*. a** Microstructure of scales in wildtype wing. Upper panel, low magnification SEM image (wing). Lower panel, high magnification SEM image (scale). Red arrows show: cr crossrib, mr microribs, r ridge, W window. **b** Knockdown experiments of His-rich *CPs* and *BmlacA2*. Compared to Control (siRNA for *GFP*), depletion of mRNAs of His-rich *CP* genes *BmorCPR152* (45% His), *BmorCPR82* (27% His), *BmorCPR57* (15% His), or *BmorCPR129* (10% His) caused scale abnormalities. Reduction of *Bmlac2A* mRNA in the pupal stage caused severe damage to scales, which seemed to encompass reduction of all His-rich *CP* mRNAs. magenta arrows, abnormal ridges; orange arrowhead, lack of microribs; white arrowheads, unexpected striations intersecting the microribs; blue arrowhead, extra window; yellow arrowheads, intertwining of cross ribs at the window region.

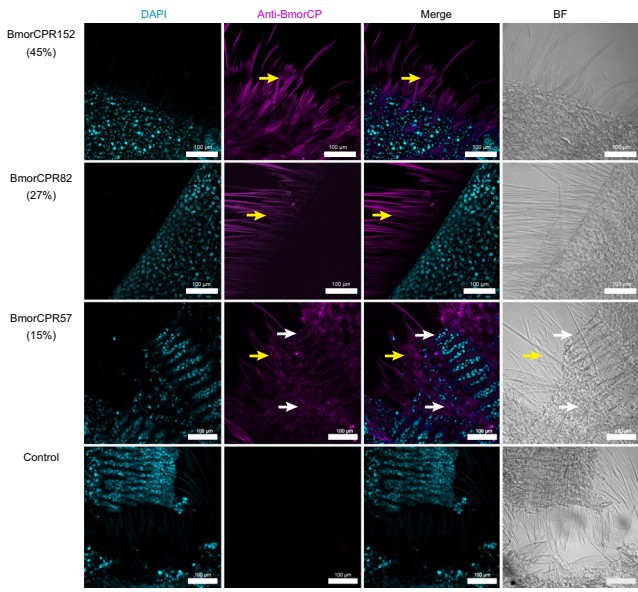

**Fig. 4 Immuno-localization of His-rich CPs in the normal wing.** BmorCPR152 (row 1) and BmorCPR82 (row 2) are mainly distributed on the scales. A very low signal of BmorCPR57 (row 3) was observed in both scales and wing. Control (row 4). white arrows, wing; yellow arrows, scales; turquoise, DAPI; magenta, Goat Anti-rabbit IgG /Alexa Fluor 555; BF, bright field (phase contrast). The colors of images were processed with Photoshop CS software (Adobe). Scale bar = 100 μm.

and *BmorCPR57* or were curved after reduction of *BmorCPR82*, *BmorCPR129* and *Bmlac2A* transcripts (Supplementary Fig. 7b). Altogether these results suggested the His-rich CPs identified in wing scales also play important roles in the formation of cuticle structures in other organs.

**Crosslinking with a His-rich CP in vitro.** Crosslinking experiments of BmorCPR152, the CP with the highest His-content found in our study, were carried out in vitro to examine whether it can be cross-linked by NADA and NBAD derivatives with

laccase2. The crosslinking reaction was performed according to previous reports[57,58] and the reaction products were resolved on SDS-PAGE gels (Fig. 5a). The reaction of BmorCPR152 with NBAD/NADA and Bmlac2A formed dimer, trimer and tetramer at 0.5 h after the reaction started, and the higher polymer bands appeared at 1 h. However, using NADA as substrate presented weaker oligomer bands and rare polymer bands compared with NBAD, and importantly, monomers were deleted faster and far weaker at 1 h and almost used up at 2 h with NADA (Fig. 5a). This suggested that polymer formation with NADA was more rapid than with NBAD and led to polymers too large to penetrate into the running gel. That NADA showed very rare high polymer bands compared with NBAD may be because both samples were centrifuged at 12,000 rpm for 5 min after the reaction, which removed large suspended protein complexes from the NADA supernatant before loading onto the SDS-PAGE gel.

The amino acid analysis of BmorCPR152 showed that HHHH/XHHH/XHH (X = H/D/G/S) repeats were evenly spread throughout the His-region (Supplementary Fig. 8). Since the imidazole group of a histidine residue is highly nucleophilic, it is a candidate for binding to α, β-dehydro-NADA/NBAD-para-quinone methide[49] (Fig. 5b). Although other amino acids like lysine (Lys), alanine (Ala), and tyrosine (Tyr) could also be involved in crosslinking, the content of those amino acids in BmorCPR152 was very low (Lys: 0.43%, Ala: 2.5%, Tyr: 2.5%). It would also be worthwhile to study the relationship between crosslinking rate and His-content of different CPs.

**Films synthesized by a His-rich CP.** Unexpectedly, films appeared on the surface of the solution after eight hours of crosslinking using the 45% His-rich CP BmorCPR152 with Bmlac2A and NADA/NBAD. The films were easily crumpled and partly folded (yellow arrows in Fig. 5c, d) when hit with droplets of PBS from a pipette. Films synthesized using NADA as the substrate (Fig. 5c) appeared white after transfer to a Petri dish for observation, whereas the films synthesized with NBAD (Fig. 5d) were golden with the occurrence of light interference in the winkled area (white arrow in Fig. 5d). As Andersen suggested, a CP will link to the α, β carbon of a dehydro-NADA methide side chain, resulting in the formation of a colorless product, which we

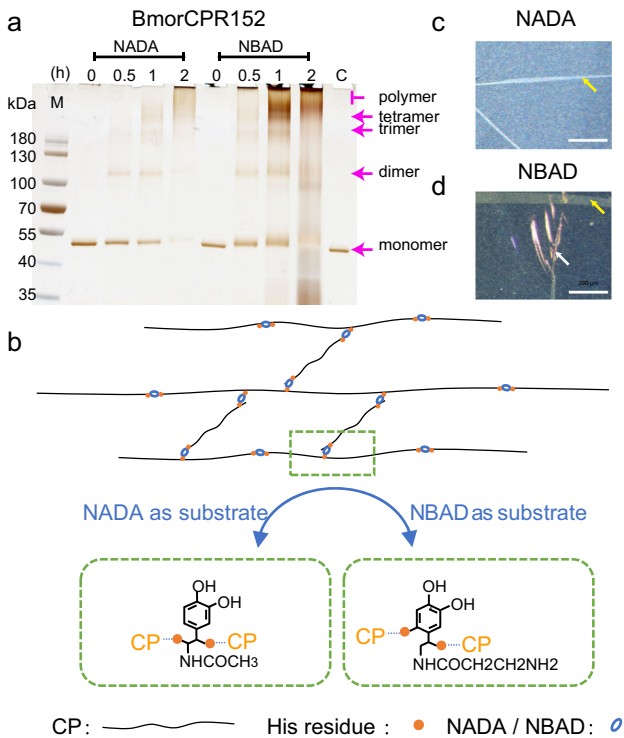

**Fig. 5 Crosslinking of BmorCPR152 with Bmlac2A and NADA/NBAD in vitro. a** Products of in vitro crosslinking reactions of BmorCPR152, examined by SDS-Page gel electrophoresis. **b** Schematic diagram of the crosslinking of CPs. Green box, interaction between histidine residues (orange dots) and NADA/NBAD (blue ovals). **c, d** Films synthesized using NADA or NBAD as substrate. Part of the films removed from the reaction buffer and observed microscopically; Scale bar = 200 μm. **c** A white film synthesized using NADA as the substrate; **d** A golden film synthesized when NADA was replaced by NBAD. Yellow arrows, wrinkles on films; white arrow, light interference occurring in the wrinkled area.

found in films synthesized using NADA; however, dehydro-NBAD methide is suggested to form links to CPs via the aromatic ring positions and β-positions of side chains, which lead to yellow or dark brown cuticle[45,49,59], as we observed using NBAD. Furthermore, no film was synthesized in the absence of Bmlac2A or NADA or NBAD (Supplementary Fig. 9), again consistent with previous reports[45,49,59]. Further study is needed to clarify the binding sites between the histidine imidazole groups of CPs and dehydro-NADA/NBAD-para-quinone methide.

The film produced by BmorCPR152 with NBAD or NADA and Bmlac2A creased easily when removed from the reaction buffer with forceps, whereas it unfolded when returned to the buffer, suggesting that it was very soft. The elastic modulus value was estimated to be 0.17–0.20 GPa by AFM measurements (Supplementary Fig. 10). These values fall into the range of 1 MPa–10 GPa for soft rubbers and polymers[60]. Although more detailed measurement of the physical properties of such cuticular films is needed, the fact that they can be formed readily in vitro and in the absence of chitin points to a potential industrial application of cuticular proteins as new biomaterials.

## Conclusion

Scales are a common attribute of Lepidoptera; however, no comprehensive information about the association of specific cuticle protein sequences with scale structure has been available. The experiments reported here indicate the intrinsic properties of

the previously unstudied scale-forming CPs may play a central role in the final formation of the exquisite microstructure of lepidopteran scales.

In our present work, we were successful in dissolving the developing scales in pupal stage and pharate adult stages before they were cross-linked; this enabled us to identify a distinctive class of His-rich CPs in lepidopteran scales. Functional studies using RNA interference on four His-rich CP genes and Bmlac2A gave direct evidence that His-rich CPs and cuticle sclerotization play crucial and precise roles in the formation of the ridges, crossribs, and microribs of the scales.

Previous reports discussed above present details about the role of chitin and actin in scale development[25,26,41]. Importantly, none of these papers mentioned cuticular proteins. We have now shown that reduction of specific CPs can produce specific modifications of scale microstructure similar to those achieved by alteration of actins and chitin.

Importantly, our experiments established that His-rich CPs could be crosslinked with Bmlac2A and NADA/NBAD, yielding a His-rich CP film even without the involvement of other fibril-forming materials like F-actin or chitin.

Our results also represent a breakthrough for the future application of CPs as a new industrially useful biomaterial. Thus, it may be possible to design a diverse array of films and fibers made of CPs by controlling the His content, thickness, stiffness, strength, elasticity, and color during synthesis in vivo or in vitro.

## Methods

**Insects and sample preparation**. *B. mori*, strain Dazao, was provided by the State Key Laboratory of Silkworm Genome Biology, Southwest University, China. The larvae were reared on fresh mulberry leaves at 25 °C (day/night, 12/12 h). The wings were easily separated from the pupal cuticle at the seventh day after pupation (P-7), with adult eclosion occurring at day 8 or 9.

Wings were dissected out at the sixth and eighth days after pupation. After washing with PBS (pH ~7.4) scales were carefully pulled off from the wings in PBS using sharpened forceps under a ZEISS Stemi2000 digital microscope (ZEISS, Germany). The PBS with scales was centrifuged (2000 rpm for 10 min) precipitating small fragments of wing and fatbody. These were discarded. The supernatant was then centrifuged at 12,000 rpm for 10 min to precipitate scales and leave contaminating hemolymph in the supernatant.

The purified scales were dissolved by shaking overnight in 8 M Urea/100 mM DTT in PBS at 4 °C, and the supernatant was collected after centrifugation at 12,000 rpm for 30 min. The pellet was washed with 25 mM NH₄HCO₃, suspended in 50 μl SDT buffer, and boiled for 15 min, followed by treatment with an ultrasonic Processor (Power, 80 W; working time, 15 s; interval, 15 s; cycles, 10×). After centrifugation at 12,000 rpm for 30 min, the supernatant was collected. Both supernatants were combined for LC-MS/MS analysis.Thus there were four samples for LC-MS-MS, two replicates from P-6 and two from P-8. A flow chart of this procedure is in Supplementary Fig. 1.

Since we could only see a negligibly small amount of pellet after this treatment, we determined that scales from P-6 pupal wings were almost completely dissolved in the buffer. Scale samples from P-8 were incompletely dissolved after the same treatment. We could not collect scales before P-6, since the wing broke easily and scales disintegrated in the PBS buffer.

**Microscopic analyses**. The development of scale shape and size was observed using an OLYMPUS TH-200 digital microscope with accompanying software, Cellsens standard (OLYMPUS, Japan). Images of the whole wing and His-rich films were taken with a ZEISS Stemi2000 digital microscope (ZEISS, Germany). Detection of the fluorescence signal was carried out with a Zeiss LSM880 confocal microscope (ZEISS, Germany). Two days after moths emerged from siRNA-injected pupae, their dried wings, head, compound eye, mandibles and leg were removed and coated with Au (MCI000, Japan) for SEM using a Hitachi SU3500 SEM (Japan). For each repeat, we randomly picked three individuals from ten eclosed moths per siRNA gene treatment for phenotypic observation. Twenty scales from each wing and ten scales from each body and leg were checked. The compound eyes and mandibles of each individual were also checked.

**Quantitative reverse transcription PCR**. Whole wings were dissected and the pupal cuticle removed on the 4th, 6th, and 8th day after pupation and placed into TRIzol (Invitrogen, USA) after washing with PBS three times. RNA was isolated from wings with TRIzol (Invitrogen, USA) following the manufacturer's instructions and treated with Recombinant DNase I (RNase-free) (Takara Bio, China) to

remove contaminating genomic DNA. Total RNA (2.5 μg) from each sample was used to construct a cDNA library using the instructions for the Reverse-Transcription Reaction Kit (Takara Bio, China). RT-qPCR was performed using SYBR Premix ExTaq II (Takara Bio, China) on an ABI7500 (ABI, USA) with the following program: 95 °C for 3 min followed by 40 cycles of 95 °C for 5 s, and 60 °C for 30 s, with a final cycle of 95 °C for 15 s, 60 °C for 20 s, and 95 °C for 15 s. The expression level of CPs was normalized to the control reference gene SWU22934 (translation initiation factor 4 A)[61]. RT-qPCR of each selected gene was repeated three times. Primers are listed in Supplementary Table 1. A dissociation curve analysis was performed for all primer sets to insure each experimental sample yielded a single sharp peak at the melting temperature of the amplicon.

**Liquid chromatograph-tandem mass spectrometry (LC-MS/MS)**. The protein concentration of the dissolved scales was determined with a BCA protein analysis kit (Beyotime Biotechnology, Shanghai, China). The Filter Aided Sample Preparation method[62] was employed to digest the proteins (1 μg trypsin/per 50 μg protein, incubated at 37 °C for 20 h), and produce a highly concentrated sample without any insoluble material. Mass spectrometric analysis Q-Exactive (Thermo Finnigan, San Jose, CA, USA) was performed for separated tryptic peptides which were resolved in a Thermo Scientific EASY-nLC 1000 column (Thermo Finnigan). Spectra were searched against a database based on protein information deduced from silkworm Gene sets A and B downloaded from KAIKObase[50] (http://sgp.dna. affrc.go.jp/ComprehensiveGeneSet/) using MaxQuant (Version 1.5.3.17) software, with a false discovery rate ≤ 0.01 (Shanghai Applied Protein Technology Co. Ltd). The comparison of protein abundance was based on the value iBAQ. All proteins identified from four samples are listed in Supplementary Data 1–4. Proteins were considered reliable if they satisfied the following criteria: (1) more than 2 peptides were identified; (2) the protein was detected in both independent replicates for each time point where it occurred.

**Antibody staining**. Peptide synthesis and polyclonal antibody production in rabbits were carried out by Zoonbio Biotechnology Co. (Nanjing, China) and Wuhan Genecreate Biological Engineering Co. (China), and the antibodies were stored at −20 °C before use. The design of antigenic peptides (Supplementary Fig. 11) and their specificity was confirmed by Western blotting using wing scales dissolved on P-6 and P-8 (Supplementary Fig. 12). For tissue immunolocalization, pupal wings were dissected at P-6 as previously described, fixed in 4% paraformaldehyde for 4 h at 4 °C, and the cell nuclei were stained with DAPI (Beyotime Biotechnology, China). The primary and second antibodies were used at a dilution of 1:1000, and tissues were washed with PBST (0.01 M PBS, 0.01%(V/V) Triton X-100) more than three times after each incubation.

**RNAi on His-rich CP genes and Bmlac2A**. siRNA sequences (Supplementary Table 2) were based on candidate genes annotated in KAIKObase using GENETYX (Software Development Co., Tokyo, Japan). Chemically synthesized double-stranded siRNAs were purchased from Takara Bio Inc. (Dalian, China) and stored at −80 °C. Two siRNAs designed against different regions of each target gene were mixed equally before use[63,64]. Injection time was determined based on the expression profile of His-rich CP genes in the wing during the pupal stage and pharate adult stage (Supplementary Fig. 13). Because transcripts for all genes examined for RNAi treatment were at a maximum level after P-4, in these experiments siRNAs were injected into pupae twice, once at P-3 and again at P-5. About 6.6 μg (20 μl × 0.33 μg/μl) of siRNAs were injected into each abdominal cavity with a glass capillary needle inserted into the third spiracle on the left side. The RNAi experiments were repeated three times using 13 pupae for each set of injections of each gene. The injected pupae were put back into their original cocoons and incubated at 25 °C. Twelve hours after the second siRNA injection, wings were dissected from three pupae that were randomly chosen from each replicate to determine the efficiency of RNAi. Three of the remaining ten injected pupae of each set were used for the observation of phenotypic difference after adult eclosion.

**Cross linking experiments with His-rich CPs in vitro**. Signal peptide coding sequences of BmorCPR152 were predicted by SignalP 5.0 (http://www.cbs.dtu.dk/services/SignalP/)[65] and removed from all CDS before using them to construct recombinant prokaryotic expression vectors by insertion into a pET-SUMO vector (provided by Wuhan Genecreate Biological Engineering Co., China), followed by expression and purification using a Genecreate Kit (Wuhan Genecreate Biological Engineering Co., China). SUMO Protease (Solarbio, China) was used to remove the SUMO-tag from the recombinant BmorCPR152 protein.

NADA and NBAD were provided by Dr. Tsunaki Asano after preparation according to Jerhot et al.[66] and Yamasaki et al.[67], respectively. Bmlac2A was purified from the abdominal cuticles of pharate pupae[54,68]. A trypsin concentration of 0.25μg/μl was used to activate laccase2 at 37 °C for 1 h before use followed by inhibition of trypsin activity with a protease inhibitor cocktail (MCE, USA)[67].

Crosslinking was carried out in 10 μl containing 1 μg/μl CP, 2 μl 0.5 M KPO₄ (pH 6.0), 2 μl 10 mM NADA or NBAD, 2 μl Bmlac2A, 2 μl protease inhibitor (10× dilution) and 2μl water at 30 °C and reaction products were collected at 0, 0.5, 1, and 2 h. Products were resolved on 4–12% (W/V) SDS-PAGE gels (Genscript, China) followed by silver staining (Beyotime Biotechnology, China) to visualize the bands.

**Measurement of physical properties of in vitro cross-linked films**. The films synthesized in vitro with the 45% His-rich CPs (BmorCPR152) were removed with forceps and, after washing with ddH₂O, were dried on a silicon slice (provided by Kawasaki laboratory of Hitachi High-Technologies, Japan). Their physical properties were measured with an Atomic Force Microscope (AFM5100N, Kawasaki laboratory of Hitachi High-Technologies, Japan).

**Statistics and reproducibility**. Graphpad Prism6 (Graph Pad software, New York, NY, USA) was used for analyzing and plotting the data. Statistical values are shown as means ± SEM. Mean values were compared using the Student's t-test with the following significance thresholds: *$P < 0.05$, **$P < 0.01$, and ***$P < 0.001$. ****$P < 0.0001$.

**Reporting summary**. Further information on research design is available in the Nature Research Reporting Summary linked to this article.

## Data availability
The LC-MS/MS analysis raw data have been deposited to the ProteomeXchange Consortium via the PRIDE partner repository with the dataset identifier PXD024401, PXD024365, PXD024362, and PXD024341. All other data generated or analysed during this study are included in this published article and presented in Supplementary Information and Supplementary Data files. All relevant data are available from the corresponding author K. Mita upon request (to mitakazuei@gmail.com).

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

## Acknowledgements

We are very grateful to the Kawasaki laboratory of Hitachi High-Technologies (Kawasaki, Japan) for their help in our work. This work was supported by a grant from the One Thousand Foreign Experts Recruitment Program of the Chinese Government (No. WQ 20125500074).

## Author contributions

K.M., and J.L. designed the whole project. J.L., Z.C., and K.M. contributed to the collection of the developing scales. T.A. contributed to the preparation of Bmlac2A, NADA and NBAD. J.L. and Y.X. helped with the insect rearing; J. L. and J. H. W. carried out the LC-MS/MS data analysis, J.L. did the RNAi and crosslinking experiments with the assistance of Y.X., Z.C., S.L., L.P., E.C., J.Z., W.L., and Y.Z. J.L. wrote the manuscript. K.M., M.R.G., J.H.W., Q.X., K.K.-O., N.H., X.T., P.Z., K.P.A., and K.P.G. revised the manuscript.

## Competing interests

The authors declare no competing interests.
