## [Peer Review File · Communications Biology]

Reviewers' comments:

Reviewer #1 (Remarks to the Author):

Comments on COMMSBIO-20-1548-T

The manuscript titled "Lepidopteran wing scales contain abundant cross-linked film-forming histidine-rich cuticular proteins" by Jianqiu Liu et al. describes the results of wing scale protein composition analysis in the silkworm. They identified a distinctive class of histidine rich CPs by LC-MS/MS and analyzed by RNAi the function of some of these proteins in constructing the microstructure of the scale surface. They also synthesized cross-linked films in vitro. The study is interesting and well designed and performed. However, some concerns and issues need to be addressed.

Specific comments:

1. Figure 1C: Are the proteins isolated from the supernatant of the 8M urea/100mM DTT treatment the same or similar to those isolated from the pellets after the treatment of 8M urea/100mM DTT followed by suspension in SDT buffer and ultrasonic treatment? The authors provided a list for P6 and P8 proteins. I would suggest the authors to give two lists of the proteins isolated from the different processes so that it would be known more accurately that what proteins are probably involved in stabilization of the structure.
2. Figure 1D: Are the numbers of CPs the ones from the supernatant of the treatment of 8M urea/100mM DTT, or the ones from the pellets after the treatment of 8M urea/100mM DTT followed by suspension in SDT buffer and ultrasonic treatment, or the ones of the first plus second treatment? Are these numbers average numbers? How many replicates they were from? Are they statistically significantly different or not?
3. Figure 2: The authors injected siRNA at P3 and P5. When did they detect the transcript expression after siRNA injection (Figure 2C)? When did they examine the phenotype changes (Figure 2A and B)? Did they check CP protein expression? They did check the locations of these proteins in normal wings (Figure 3A), but did not check the protein expression after injection? In the siRNA treatments, the suppression of the transcript expression of all these His-rich CP genes reached more than 50%, but the genotypes of these treatment revealed the microstructure damage is not highly and closely associated with the observed changes in the transcript level. How to explain this?
4. Supplementary Figure 7: The authors showed the abnormality of the wing development after siRNA of CPs and Bmlac2. They "used 13 pupae for each set of injections of each gene." How many animals showed the wing abnormality in each of siRNA treatments? How many replicates were done? Do they have the original data and statistic data?
5. Figure 4A: What is the predicted molecular mass of BmorCPR152? What is the lane C for in the figure? The authors mentioned that "the reaction of BmorCPR152 with NBAD and Bmlac2 formed dimer, trimer, and tetramer 30 min after the reaction started, and higher polymer bands appeared at six hours. By contrast, when NADA was used as substrate, weaker oligomer bands and very rare polymer bands were observed; but importantly the monomer band was far weaker at 2 hr and almost absent at 6 hr. This suggests that polymer formation with NADA was more rapid than with NBAD and led to polymers too large to penetrate into the running gel." If it is true, the large polymers should be observed in the sample loading wells of the gel, as it is seen in NBAD 6h, but it is not. How to explain this? No obvious and solid evidence to show that polymer formation with NADA was more rapidly than that with NBAD. Also, there is no obvious and solid evidence to show larger polymers are formed in NADA than in NBAD. Because Bmlac2 was added to the reaction system, it may catalyze the formation of NADA-ortho-quinone from NADA, but not catalyze the reaction of NBAD. This may result in more NADA sclerotization. But this is not because NADA or NBAD itself.
6. Figure 4B: this figure provides no information for the polymerization. Instead, I would suggest replacing it with a diagram of structural interaction between the protein and NADA/NBAD. What is the relationship between the His-rich region and chitin binding domain? Are the His-rich region for polymerization and the chitin-binding domain for chitin binding? Any experimental evidence? Such as amino acid residue mutation?
7. Figure 4C and D: Is there any difference in the physical properties between the NADA and NBAD

films? Considering the potential application, no chitin was added in this experiment. However, if chitin is added, the crosslinking and physical properties of the films would be improved or not? In this experiment, Bmlac2 was purified from the abdominal cuticles of pharate pupae of the silkworm. How to purify? No method is provided in the manuscript. Did the authors try different concentration combinations of the crosslinking with BmorCPR152, Bmlac2 and NADA or NBAD? Why did the authors run the crosslinking experiment in acidic pH 6.0? What is the pI of BmorCPR152?

8. Table 1: The authors list "the most abundant proteins" in the base of iBAQ values. This is no problem within a sample. However, if they want to compare the abundance between the samples, it would be better to use MaxLFQ (Reference: Cox J et al. Accurate proteome-wide label-free quantification by delayed normalization and maximal peptide ratio extraction, termed MaxLFQ. *Molecular & Cellular Proteomics* Mcp, 2014, 13(9):2513).

9. Table 1: The authors listed the most abundant proteins isolated from P6 and P8. Again, were these proteins isolated from the supernatant of the 8M urea/100mM DTT treatment or from those isolated from the pellets after the treatment of 8M urea/100mM DTT followed by suspension in SDT buffer and ultrasonic treatment? This would be important to learn what CPs are more tightly bound with chitin. In addition, as mentioned by the authors, at P8, the cuticle became more insoluble than at P6. If the data derived from P6 is used to subtract the data from P8, if they can estimate those proteins insoluble in P8?

10. This manuscript identified many His-rich CPs and believed they are involved in the cuticle formation in the wing scales. Are these proteins scale-specific or are they also involved in wing epidermis formation? As compared to other CPs, such as glycine-rich, proline-rich, and cysteine-rich CPs, if the His-rich CPs play different or more critical roles in the formation of scale cuticle? What is the structural and functional relationship between the His-rich region and the R& R consensus?

Reviewer #2 (Remarks to the Author):

This is a great paper that begins to fill the gap between the molecular biology and physiology of butterfly wing pigmentation. All the findings are new and exciting. The link between the gene knock-out studies and biochemistry is especially impressive. The only issue I have with the paper is the historical context in which it is set. Specifically it states that Montero et al. were the first to implicate the link between colouration and sclerotization in wing scales- but this ignores the work of B. Koch and others from the Tiger Swallowtail which first described the link between sclerotisation and final scale colour and importantly also provided a theoretical framework within which it can be examined. I therefore recommend the inclusion of the Koch et al. papers in the introduction and discussion. Otherwise its brilliant! Congratulations to the authors.

Reviewer #3 (Remarks to the Author):

This manuscript describes identification and characterization of cuticle proteins (CPs) in scales of the silkworm, and cross-linking experiments for CPs. The author developed the method for dissolving the CPs from scales, analyzed the functional roles of some cuticular proteins (His-rich proteins) by knockdown experiment by siRNA injection, and produced film-like structure by in vitro crosslinking of His-rich CPs. Since the CPs in scales of Lepidoptera are largely unknown, and thus the findings are novel and interesting, and helpful to understand the functional roles of His-rich CPs in scale formation. However, I feel that some points should be clarified before publication. I have several questions and comments as follows.

Comments

1. The author claimed the extraction method of CPs from scales (wings) as a breakthrough approach (Line 41). I am not sure whether they improved the method (such as buffer or some procedure) itself or the developmental stage (P6, P8) when the wing cuticle is soft was effective for extraction. Please clarify what is breakthrough.
2. All proteins included in the wings at the P6 or P8 stages could be extracted equally by this method? How much % proteins are yielded? The authors commented that some proteins such as

ebony were not included in the extracted proteins due to its low expression. To use this logic for expression level of the specific protein, it is important to explain this.

3. Table 1; Are the His-rich CPs specifically expressed in scales? Or expressed also in other tissues such as larval cuticles? I think it better to explain whether CPs listed in tables are specific to pupal stages and specific to scale epidermis, or not.

4. Fig. 2; In the RNAi analyses, the authors used 13 pupae for phenotypic observation. It is better to show how many larvae treated with siRNA showed the phenotype shown in Fig. 2.

5. Fig. 2; By injection of siRNA in pupal body (abdominal part), all parts of body other than wings should be affected by knockdown of gene of interest. Especially, knockdown of laccase II working on whole body sclerotization should affect not only scale structure but also other morphology (such as scaly hair in the body, antenna, eye and etc). Did you see the RNAi effect on other parts than wings.

6. Fig. 3; The location of CPR57 not only in scales but also in wing blades (shown by arrows in merged photo) is not clear for me.

7. Fig. 4; The author claimed that NADA substrates more effectively polymerize CPR152, because a monomer band was less observed at 6h than at 2 h for reaction. In case of NBAD, the polymerized band increased according to the reaction time, and the author concluded that polymer formation with NADA was more rapid than with NBAD and a larger polymer with NADA could not penetrate the gel. However, at 0.5 h or 1 h reaction, we can see much monomer without polymerized bands, which seems strange. The authors should show the negative control without NADA or NBAD, and 0 min reaction.

Response to reviewers' comments

Reviewer #1:

1. Figure 1C: Are the proteins isolated from the supernatant of the 8M urea/100mM DTT treatment the same or similar to those isolated from the pellets after the treatment of 8M urea/100mM DTT followed by suspension in SDT buffer and ultrasonic treatment? The authors provided a list for P6 and P8 proteins. I would suggest the authors to give two lists of the proteins isolated from the different processes so that it would be known more accurately that what proteins are probably involved in stabilization of the structure.

Reply: We wanted to know which CP(s) participate in scale formation. Since proteins cannot be resolved by ultrasonic treatment after crosslinking, we combined both supernatants as a single "supernatant". We stated this in Lines 78-82: "Both supernatants were combined for LC-MS/MS analysis after an aliquot (20 ug) was removed for electrophoresis. Since the electrophoretic banding patterns were so similar from the first and second extracts (Fig 1C), they were combined before LC-MS/MS. Thus there were four samples for LC-MS/MS, two replicates from P-6 and two from P-8." So there are no data to allow us to compare the proteins isolated from the different processes.

2. Figure 1D: Are the numbers of CPs the ones from the supernatant of the treatment of 8M urea/100mM DTT, or the ones from the pellets after the treatment of 8M urea/100mM DTT followed by suspension in SDT buffer and ultrasonic treatment, or the ones of the first plus second treatment? Are these numbers average numbers? How many replicates they were from? Are they statistically significantly different or not?

Reply: The numbers in Fig. 1D were from the first plus the second treatment, as noted above. We made two replicates for each time point. As we mentioned in section 3.3, only “reliable” proteins were used for the analysis in this report. According to our description in section 2.4, the proteins were considered reliable if they satisfied the following criteria: (1) more than 2 peptides were identified; (2) the protein was detected in both independent replicates for each time point where it occurred.

3. Figure 2: The authors injected siRNA at P3 and P5. When did they detect the transcript expression after siRNA injection (Figure 2C)? When did they examine the phenotype changes (Figure 2A and B)? Did they check CP protein expression? They did check the locations of these proteins in normal wings (Figure 3A), but did not check the protein expression after injection? In the siRNA treatments, the suppression of the transcript expression of all these His-rich CP genes reached more than 50%, but the genotypes of these treatment revealed the microstructure damage is not highly and closely associated with the observed changes in the transcript level. How to explain this?

Reply: As we mentioned in Lines 162-165 “Twelve hours after the second siRNA injection, wings were dissected from three pupae that were randomly chosen from each replicate to determine the efficiency of RNAi.” We did not check the CP protein expression in scales after injection, since it is very difficult to collect enough scales from such a small number (less than 10) of injected pupae. Two days after moths emerged from siRNA-injected pupae, their dried wings, head, compound eye, mandibles and leg were removed and coated with Au (MCI000, Japan) for scanning electron microscopy (SEM) using a Hitachi SU3500 SEM (Japan). For each repeat, we randomly picked 3

individuals from 10 eclosed moths per siRNA gene treatment for phenotypic observation. Twenty scales from each wing and 10 scales from each body and leg were checked. The compound eyes and mandibles of each individual were also checked (described in section 2.2, lines 98-101). We added the statistical analysis data to Fig. 2B (left panel) and related description in section 3.5 as follows: “Reduction of the mRNAs of the 4 His-rich *CPs* with different His-content (45%, 27%, 15% and 10%) and *Bmlac2A* led to abnormal microstructures in the wing scales of over half of the examined scales (details in Fig. 2B left panel).” (lines 343-346).” We think the depletion of transcripts by RNAi treatment correlates with the abnormal ratios that appeared. The most important findings are that the microstructures of the scales were significantly damaged by RNAi and depletion of different transcripts resulted in different abnormalities.

4. Supplementary Figure 7: The authors showed the abnormality of the wing development after siRNA of *CPs* and *Bmlac2*. They “used 13 pupae for each set of injections of each gene.” How many animals showed the wing abnormality in each of siRNA treatments? How many replicates were done? Do they have the original data and statistic data?

Reply: We repeated the RNAi experiment three times. We added a new figure (Fig. 2 right panel) to show the wing abnormalities after each siRNA treatment. We revised the related sentence in section 3.5 (lines 343-346) as follows: “Reduction of the mRNAs of the 4 His-rich *CPs* with different His-content (45%, 27%, 15% and 10%) and *Bmlac2A* led to abnormal microstructures in the wing scales of over half of the examined scales

(details in Fig. 2B left panel).” In addition, abnormalities in wing structure were found in knockdown experiments of 15% and 10% His-rich CPs and Bmlac2A whereas there was no apparent effect on wing structure from knockdown of the 45% and 27% His-rich CPs (as shown in Fig. 2B right panel).”

5. Figure 4A: What is the predicted molecular mass of BmorCPR152? What is the lane C for in the figure? The authors mentioned that “the reaction of BmorCPR152 with NBAD and Bmlac2 formed dimer, trimer, and tetramer 30 min after the reaction started, and higher polymer bands appeared at six hours. By contrast, when NADA was used as substrate, weaker oligomer bands and very rare polymer bands were observed; but importantly the monomer band was far weaker at 2hr and almost absent at 6 hr. This suggests that polymer formation with NADA was more rapid than with NBAD and led to polymers too large to penetrate into the running gel.” If it is true, the large polymers should be observed in the sample loading wells of the gel, as it is seen in NBAD 6h, but it is not. How to explain this? No obvious and solid evidence to show that polymer formation with NADA was more rapidly than that with NBAD. Also, there is no obvious and solid evidence to show larger polymers are formed in NADA than in NBAD.

Because Bmlac2 was added to the reaction system, it may catalyze the formation of NADA-ortho-quinone from NADA, but not catalyze the reaction of NBAD. This may result in more NADA sclerotization. But this is not because NADA or NBAD itself.

Reply: The predicted molecular weight of BmorCPR152 (without signal peptide) is 28kD.

Lane C is the control without NADA or NBAD

Thank you for a possible alternative explanation of why when NADA was used as

substrate, weaker oligomer bands and very rare polymer bands were observed. Based on Comment 7 of Reviewer 3 “The authors should show the negative control without NADA or NBAD, and 0 min reaction”, we repeated the crosslinking experiment with the same reaction conditions except for adding a 0 min reaction and replaced Fig. 5A with a gel photo of the new crosslinking reaction. We changed the related description (lines 397-407 in section 3.6) to explain the different gel patterns in the NADA vs NBAD reactions as follows “The reaction of BmorCPR152 with NBAD/ NADA and Bmlac2A formed dimer, trimer and tetramer at 0.5h after the reaction started, and the higher polymer bands appeared at 1h. However when NADA was used as substrate, NADA presented weaker oligomer bands and rare polymer bands compared with NBAD, but importantly deleted monomers faster than NBAD, which were far weaker at 1 hr and almost used up at 2 hr (Fig. 5A). This suggested that polymer formation with NADA was more rapid than NBAD and led to polymers too large to penetrate into the running gel. That NADA showed very rare high polymer bands compared with NBAD may be because samples were centrifuged at 12000rpm for 5 min after the reaction, which removed the large suspended protein complexes from the supernatant loaded onto the SDS-PAGE gel.”

6. Figure 4B: this figure provides no information for the polymerization. Instead, I would suggest replacing it with a diagram of structural interaction between the protein and NADA/NBAD. What is the relationship between the His-rich region and chitin binding domain? Are the His-rich region for polymerization and the chitin-binding domain for chitin binding? Any experimental evidence? Such as amino acid residue mutation?

Reply: We revised the diagram in Fig.5B to show the interaction between the protein and NADA/NBAD. Rebers and Willis (2001, ref # 69) were the first to establish that the R&R Consensus region was both necessary and sufficient for chitin binding. Iconomidou et al. 2005 published a homology model for RR-2 proteins showing that the His residues in the Consensus region were in positions where they could participate in cross-links (ref # 43). We propose that the His-rich regions are important for polymerization between proteins through dehydro-NADA/NBAD, which was indicated by the results of crosslinking experiments described in the manuscript (section 3.6). We are now performing additional experiments using the His-region to confirm its role in the polymerization of proteins and will publish those results separately in detail.

7. Figure 4C and D: Is there any difference in the physical properties between the NADA and NBAD films? Considering the potential application, no chitin was added in this experiment. However, if chitin is added, the crosslinking and physical properties of the films would be improved or not? In this experiment, Bmlac2 was purified from the abdominal cuticles of pharate pupae of the silkworm. How to purify? No method is provided in the manuscript. Did the authors try different concentration combinations of the crosslinking with BmorCPR152, Bmlac2 and NADA or NBAD? Why did the authors run the crosslinking experiment in acidic pH 6.0? What is the pI of BmorCPR152?

Reply: Unfortunately, we did not compare the difference in the physical properties between the NADA and NBAD films in this work. To consider future applications of His-rich CP films, we are doing experiments to determine how the physical properties of the films and the crosslinking rate change using different concentrations of NADA/NBAD,

laccase2 and CPs with different His content.

We purified Bmlac2A using a published method which we reported in the manuscript (line 175, reference 37, Yatsu & Asano, 2009).

One of the novel findings of this paper is that films can be produced without chitin. No one knows the form chitin takes when it interacts with cuticular proteins *in vivo*, i.e., the length of the chitin chains or even if it is chitin or deacetylated chitin. Nor do we know anything about the properties of the extracellular assembly zone where chitin and proteins interact. The addition of chitin would indeed be interesting, and should be a follow-up study to continue exploring potential applications, but is not essential for the results we present. Indeed, what is novel about our results was that films would form in the absence of chitin.

Regarding the reaction conditions: The predicted PI of BmorCPR152 is 6.5; using an acidic buffer condition at pH 6.0 would increase the nucleophilicity of the histidine imidazole group(s). Determination of the specific activity of Bmlac2A and crosslinking experiments were performed in 0.1 M potassium phosphate buffer pH 6.0 - 6.5 (as indicated in references 38, 39 which are cited in Materials and Methods).

8. Table 1: The authors list “the most abundant proteins” in the base of iBAQ values.

This is no problem within a sample. However, if they want to compare the abundance between the samples, it would be better to use MaxLFQ (Reference: Cox J et al.

Accurate proteome-wide label-free quantification by delayed normalization and maximal peptide ratio extraction, termed MaxLFQ. *Molecular & Cellular Proteomics* Mcp, 2014, 13(9): 2513).

Reply: Thank you for these suggestions. In our study, we only wanted to show which CPs or His-rich CPs were the most abundant proteins in each sample. For this reason we did not compare the abundance of each protein between the samples.

9. Table 1: The authors listed the most abundant proteins isolated from P6 and P8. Again, were these proteins isolated from the supernatant of the 8M urea/100mM DTT treatment or from those isolated from the pellets after the treatment of 8M urea/100mM DTT followed by suspension in SDT buffer and ultrasonic treatment? This would be important to learn what CPs are more tightly bound with chitin. In addition, as mentioned by the authors, at P8, the cuticle became more insoluble than at P6. If the data derived from P6 is used to subtract the data from P8, if they can estimate those proteins insoluble in P8?

Reply: As indicated previously we used the combined material from the supernatant and pellet treated as described above for all of the samples reported in the study. The purpose of our work was to identify the proteins in the scales. We did not do a comparison of the proteins between the P6 and P8 samples to determine which proteins may be more tightly bound with chitin. Difference in solubility of specific proteins could be due to factors other than chitin-binding such as cross-linking. Indeed the recovery of so many CPR proteins, with verified chitin-binding domains, indicates that the conditions we used were sufficient to eliminate protein-chitin interactions. Studies on chitin-binding such as Rebers and Willis (2001) just used 8M urea to strip proteins from the chitin beads.

Further, although it would be important to know which proteins are actually bound with chitin, getting definitive answers to this question would be technically demanding and is beyond the scope of the aims of the present project.

10. This manuscript identified many His-rich CPs and believed they are involved in the cuticle formation in the wing scales. Are these proteins scale-specific or are they also involved in wing epidermis formation? As compared to other CPs, such as glycine-rich, proline-rich, and cysteine-rich CPs, if the His-rich CPs play different or more critical roles in the formation of scale cuticle? What is the structural and functional relationship between the His-rich region and the R&R consensus?

Reply: Whether or not these proteins are specific to scales or also found in the wing epidermis is indeed an interesting question. Unfortunately, the complete separation of wing epidermis and scales is physically impossible so far, so we cannot confirm whether the proteins we identified from scales are also involved in wing epidermis by LC-MS/MS. However, as we described in section 3.5 (lines 367-369), “After siRNA injection of the 15% and 10% His-rich CPs, we found wings were not fully expanded after the moth emerged”, which suggests some proteins identified in scales such as BmorCPR57 and BmorCPR129 may also be involved in the formation of wing epidermis. Pursuit of this question in a rigorous way will require use of techniques to localize proteins cytologically (e.g., by EM-immunolocalization), which is currently beyond the scope of this study and may be pursued in the future.

Any amino acid which contains a nucleophilic group could theoretically bind to a NADA/NBAD-para-quinone-methide (references 58 and 59); however, the His residue,

which contains the strongest nucleophilic imidazole group, is the main candidate for this reaction (references 58 and 59). So we focused on the His-rich CPs in this study.

Regarding the functional relationship between the His-rich region and the R&R consensus region, Rebers and Willis confirmed the R&R consensus region is necessary and sufficient for chitin binding (reference 68); hence, we speculate the His-rich region would be important for connecting the different CPs. So an R&R Consensus and His-rich region could cooperatively regulate the physical properties of CP structures, which would be determined by interactions between CP-Chitin and protein-protein links through a His-rich domain and NADA/NBAD. Because this model is highly speculative and does not yet have direct physical or chemical evidence we did not include a discussion of it in the manuscript.

Reviewer #2:

This is a great paper that begins to fill the gap between the molecular biology and physiology of butterfly wing pigmentation. All the findings are new and exciting. The link between the gene knock-out studies and biochemistry is especially impressive. The only issue I have with the paper is the historical context in which it is set. Specifically it states that Montero et al. were the first to implicate the link between colouration and sclerotization in wing scales- but this ignores the work of B. Koch and others from the Tiger Swallowtail which first described the link between sclerotisation and final scale colour and importantly also provided a theoretical framework within which it can be examined. I therefore recommend the inclusion of the Koch et al. papers in the introduction and discussion. Otherwise its brilliant! Congratulations to the authors.

Reply: Thank you for your positive comments and suggestion. We have cited the following related references in the introduction:

20 Koch, P. B. et al. Regulation of dopa decarboxylase expression during colour pattern formation in wild-type and melanic tiger swallowtail butterflies.

Development **125**, 2303–2313 (1998).

22. Koch, P. B., Lorenz, U., Brakefield, P. M & ffrench-Constant, R. H. Butterfly wing pattern mutants: developmental heterochrony and co-ordinately regulated phenotypes. *Dev Genes Evol* **210**, 536-44 (2000).

23. Koch, P. B., Behnecke, B. & ffrench-Constant, R. H. The molecular basis of melanism and mimicry in a swallowtail butterfly. *Curr Biol* **10**, 591-4 (2000).

Reviewer #3

1. The author claimed the extraction method of CPs from scales (wings) as a breakthrough approach (Line 41). I am not sure whether they improved the method (such as buffer or some procedure) itself or the developmental stage (P6, P8) when the wing cuticle is soft was effective for extraction. Please clarify what is breakthrough.

Reply: Our breakthrough was the examination of developing scales in which proteins were not yet cross-linked. So far molecular studies have focused only on fully sclerotic mature scales from adults, in which most of the proteins had been rendered insoluble. This work for the first time identified cuticular proteins contributing to scale microstructure and elucidated their functions in scale microstructure formation. We added a sentence to the Introduction to describe this aspect of the study more clearly (Lines 47-50: "In this study, we succeeded in identifying many CPs and other classes of

proteins by collecting scales from developing wings before crosslinking occurred and dissolving them in a buffer containing urea, a well-established method for studying cuticle CPs but a breakthrough approach for the molecular study of wing scales.”)

2. All proteins included in the wings at the P6 or P8 stages could be extracted equally by this method? How much % proteins are yielded? The authors commented that some proteins such as ebony were not included in the extracted proteins due to its low expression. To use this logic for expression level of the specific protein, it is important to explain this.

Reply: We do not think all proteins present in the wings at P-6 and P-8 can be extracted equally. We described this point in section 3.2: “These results showed that scales of P-8 became much more difficult to dissolve compared to P-6 scales, which was consistent with our morphological observations indicating that the scales started to stabilize around P-6. As the proteins and other components were stabilized, the number and amount of proteins extractable from scales were also expected to be reduced.” Otherwise, since the collection of scales in pupal stages is very difficult (see below), we think the amount of starting material needed to obtain accurate data on the protein yield at P6 and P8 stages is not feasible with these hand-collected scales.

As the scales are very tiny, the collection and purification are very difficult and hard work. We think it is not technically possible to determine what percent of the total protein is extracted in this work.

Absence of ebony: We are sorry that we cannot understand what this comment means. The manuscript states that ebony was not recovered from scales although it

was reported in silkworm (larval) epidermis by Futahashi et al. Our manuscript did not write anything about its low expression as an explanation of the failure to find it during formation of adult cuticle.

3. Table 1; Are the His-rich CPs specifically expressed in scales? Or expressed also in other tissues such as larval cuticles? I think it better to explain whether CPs listed in tables are specific to pupal stages and specific to scale epidermis, or not.

Reply: Thank you for your suggestion. Judging from the EST database from KAIKObase, the four His-rich PCs that we mentioned in our MS are mainly expressed in the pupal stage, and the expression levels of those genes were much higher in wing than other tissues. We added the EST data for the top 25 genes in Supplementary Table 2, where column H now lists the transcript level of those genes in various tissues at different stages.

4. Fig. 2; In the RNAi analyses, the authors used 13 pupae for phenotypic observation. It is better to show how many larvae treated with siRNA showed the phenotype shown in Fig. 2.

Reply: Thank you for your valuable comment. First we would like to explain our RNAi experiment as described in section 2.6. We injected 13 pupae of each set for each gene, among which 3 pupae were used to determine the efficiency of RNAi at 12 hrs after last injection. The remaining 10 injected pupae of each set were used for phenotypic observation. We added one sentence at the last of section 2.6 (Line 162-165) to avoid misunderstanding for readers. In response to this comment we also added

a new figure (Fig. 2), which reports the statistical analysis of percent abnormality appearing in wing (right panel) and wing scales (left panel) after RNAi treatment. The related description is revised in section 3.5 (lines 343-346) of the manuscript as follows: “Reduction of the mRNAs of the 4 His-rich *CPs* with different His-content (45%, 27%, 15% and 10%) and *Bmlac2A* led to abnormal microstructures in the wing scales of over half of the examined scales (details in Fig. 2B left panel).” In addition, abnormalities in wing structure were found in knockdown experiments of 15% and 10% His-rich *CPs* and *Bmlac2A* whereas there was no apparent effect on wing structure from knockdown of the 45% and 27% His-rich *CPs* (as shown in Fig. 2B right panel).”

5. Fig. 2; By injection of siRNA in pupal body (abdominal part), all parts of body other than wings should be affected by knockdown of gene of interest. Especially, knockdown of laccase II working on whole body sclerotization should affect not only scale structure but also other morphology (such as scaly hair in the body, antenna, eye and etc). Did you see the RNAi effect on other parts than wings.

Reply: Thank you for this helpful comment. We also examined body scales and leg scales as well as compound eye and mandibular bristles. We reported an RNAi effect on those structures in Supplementary Figures 8 and 9 and added the following description to the last paragraph in section 3.5 (lines 381-491): “We also examined the scales from the body and legs after RNA knockdown, which showed abnormalities similar to the microstructure of wing scales (Supplementary Figs. 8 and 9). We found severe damage to the compound eyes and mandibular bristles after interference in the expression of the His-rich *CPs* and *Bmlac2A* (Supplementary Fig. 10). Especially, the

surface of the compound eyes was shrunken by reduction of the mRNAs of *BmorCPR152* and *Bmlac2A* (supplementary Fig10A). In addition, some mandibular bristles were missing after RNAi treatment with *BmorCPR152* and *BmorCPR57* or were curved after reduction of *BmorCPR82*, *BmorCPR129* and *Bmlac2A* transcripts (supplementary Fig10B). Altogether these results indicated the His-rich CPs identified in wing scales also play important roles in the formation of cuticle structures in other organs.”

6. Fig. 3; The location of CPR57 not only in scales but also in wing blades (shown by arrows in merged photo) is not clear for me.

Reply: Thank you for this comment. We have added new arrows to Fig.3 to designate the location of CPR57, which will be helpful for readers to know its position in scale and wing blades in the image.

7. Fig. 4; The author claimed that NADA substrates more effectively polymerize CPR152, because a monomer band was less observed at 6h than at 2 h for reaction. In case of NBAD, the polymerized band increased according to the reaction time, and the author concluded that polymer formation with NADA was more rapid than with NBAD and a larger polymer with NADA could not penetrate the gel. However, at 0.5 h or 1 h reaction, we can see much monomer without polymerized bands, which seems strange. The authors should show the negative control without NADA or NBAD, and 0 min reaction.

Reply: Thank you for your astute observation. Lane C in Fig. 5A is the negative

control without NADA or NBAD. In consideration of this comment, we repeated the crosslinking experiment with the same reaction condition except for adding a 0 min reaction and replaced Fig. 5A with a gel photo of the new crosslinking reaction. We found “The reaction of BmorCPR152 with NBAD/ NADA and Bmlac2A formed dimer, trimer and tetramer at 0.5h after the reaction started, and the higher polymer bands appeared at 1h. However when NADA was used as substrate, NADA presented weaker oligomer bands and rare polymer bands compared with NBAD, but importantly deleted monomers faster than NBAD, which were far weaker at 1 hr and almost used up at 2 hr (Fig. 5A). This suggested that polymer formation with NADA was more rapid than NBAD and led to polymers too large to penetrate into the running gel. That NADA showed very rare high polymer bands compared with NBAD may be because samples were centrifuged at 12000rpm for 5 min after the reaction, which removed the large suspended protein complexes from the supernatant loaded onto the SDS-PAGE gel.” (Lines 397-407)

But, as Reviewer 1 pointed out, we don't have a clear and solid evidence to show that larger polymers are formed in NADA than in NBAD, nor is the precise mechanism of crosslinking determined, and remains for further study.

Based on our present experiments, the important finding is the His-rich CPs could be polymerized with NADA or NBAD and Bmlac2A to form the film, although the precise mechanism of the cross linking process is not yet established.

REVIEWERS' COMMENTS:

Reviewer #1 (Remarks to the Author):

The authors have addressed most of my questions and concerns and done additional experiments. I would suggest the paper to be accepted for publication. I would also like to suggest that it is critical important to explore the interactive link between scale proteins and chitin in the wing scales and hope to see their follow-up works on this.

REVIEWERS' COMMENTS:

Reviewer #1:

The authors have addressed most of my questions and concerns and done additional experiments. I would suggest the paper to be accepted for publication. I would also like to suggest that it is critical important to explore the interactive link between scale proteins and chitin in the wing scales and hope to see their follow-up works on this.

Reply: Thank you for your suggestion. Our research will provide fundamental information about the interaction between the scale proteins and chitin in future work.